# Integrated Genetic Characterization and Quantitative Risk Assessment of Cephalosporin- and Ciprofloxacin-Resistant *Salmonella* in Pork from Thailand

**DOI:** 10.3390/antibiotics14121198

**Published:** 2025-11-27

**Authors:** Thawanrut Kiatyingangsulee, Si Thu Hein, Rangsiya Prathan, Songsak Srisanga, Saharuetai Jeamsripong, Rungtip Chuanchuen

**Affiliations:** 1National Institute of Animal Health, Department of Livestock Development, Bangkok 10900, Thailand; niah4@dld.go.th; 2Department of Veterinary Public Health, Chulalongkorn University, Bangkok 10330, Thailand; 3Department of Anatomy, University of Veterinary Science, Yezin 15013, Myanmar; drsithuhein@uvs.edu.mm; 4Research Unit in Microbial Food Safety and Antimicrobial Resistance, Department of Veterinary Public Health, Faculty of Veterinary Science, Chulalongkorn University, Bangkok 10330, Thailand; rangsiya.p@chula.ac.th (R.P.); songsak.s@chula.ac.th (S.S.); saharuetai.j@chula.ac.th (S.J.); 5Center for Antimicrobial Resistance Monitoring in Food-Borne Pathogens, Faculty of Veterinary Science, Chulalongkorn University, Bangkok 10330, Thailand

**Keywords:** ciprofloxacin resistance, ESBL production, pork, QMRA, *Salmonella*

## Abstract

**Background/Objectives**: This study assessed the risk associated with third-generation cephalosporin- and fluoroquinolone-resistant *Salmonella* from pork consumption by integrating phenotypic resistance profiles with genetic data to characterize the risks and transmission pathways. **Methods**: *Salmonella* were isolated from raw pork meat samples (*n* = 793) collected from fresh markets and hypermarkets across Bangkok during 2021–2022, of which 150 were extended-spectrum β-lactamase (ESBL)-producing and 31 were fluoroquinolone-resistant isolates. Phenotypic and genotypic resistance profiles were characterized. Quantitative antimicrobial resistance risk assessment (AMR RA) was conducted using a dose–response model. **Results**: *Salmonella* spp. was detected in 42.75% of pork samples, with a higher prevalence in fresh markets (75.5%) than in hypermarket samples and with concentrations ranging from 1.3 to 180 MPN/g. Twenty-eight percent of isolates were ESBL producers, with ciprofloxacin and levofloxacin resistance observed in 5.3% and 3.0%, respectively. The *bla*_CTX-M55_ genes were located on conjugative plasmids. Whole genome sequencing revealed both vertical and horizontal gene transfer. IncHI2/N and IncC plasmids shared conserved backbones and resistance gene architectures, indicating horizontal dissemination of resistance genes. Phylogenomics suggested possible clonal transmission among pigs, pork, and humans. AMR RA estimated 88,194 annual illness cases per 100,000 people from ESBL-producing *Salmonella* and 61,877 from ciprofloxacin-resistant strain, compared with 95,328 cases predicted by QMRA from *Salmonella* contamination. Cooking pork at ≥64 °C for 3 min eliminated the risk in all scenarios. Sensitivity analysis identified initial contamination level and cooking temperature as key determinants. **Conclusions**: Raw pork meat consumption represents the highest risk, which can be mitigated by thorough cooking (>64 °C, ≥3 min), while integrating genomic data enhances AMR hazard identification, source attribution, and exposure assessment. Therefore, promoting well-cooked meat consumption and safe cooking practices, alongside the use of AMR genetic data to inform targeted interventions, is recommended.

## 1. Introduction

*Salmonella enterica* is a leading foodborne pathogen, causing over 94 million cases of salmonellosis and approximately 150,000 deaths annually worldwide [1]. *Salmonella* contamination in raw meat is a major food safety concern in developing countries, including Thailand where over 90% of the population consume pork daily, with annual consumption reaching 1.3 million tons [2,3]. Thailand’s pig production ranged from 16 to 22 million heads, and pork consumption from 1.13 to 1.49 million tons during 2020–2024 [3]. High *Salmonella* contamination in pork sold in traditional markets (40–80%) has been reported [4,5], highlighting the ongoing risk to consumers and the need to enhance pork supply chain safety. Importantly, raw pork meat consumption is popular in certain parts of Thailand as some traditional dishes such as Larb Moo Dib (raw pork salad), Luh (raw pork blood dish), and Nham (fermented raw pork) are sometimes prepared with uncooked pork. Such consuming behavior intensifies the risk of foodborne pathogens. This risk is further compounded by the emergence of antimicrobial resistance in *Salmonella*, which can disseminate across the food chain.

AMR represents a major global health challenge, causing 1.27 million deaths annually and over $100 billion in economic losses [6]. Its emergence and spread is primarily driven by antibiotic overuse and misuse in humans and livestock [7]. It was estimated that antimicrobial use in food animals accounted for 73% of global sales, with pigs consuming the most at 193 mg/PCU [8]. Extensive antibiotic use for treatment, prevention, and growth promotion in pigs could facilitate the selection and spread of AMR bacteria and genes, such as multidrug-resistant *Salmonella*, that can enter the food chain and pose serious risks to human health [9].

AMR spreads through vertical (VGT) and horizontal (HGT) gene transfer. VGT involves the inheritance of resistance from parent to daughter cells, while HGT enables the rapid AMR spread across species, often through transmissible plasmids and transposons that carry multiple resistance genes. Third-generation cephalosporins and fluoroquinolones are critically important antibiotics and among the most effective treatments for severe *Salmonella* infections with bacteremia and severe gastroenteritis. New-generation cephalosporin resistance is primarily mediated by extended-spectrum β-lactamase (ESBL) enzymes, which commonly spread via the horizontal transfer of plasmids carrying ESBL genes [10]. The global emergence of ESBL-producing *Salmonella* isolates have been increasingly reported in human, livestock, and meat products [11,12,13]. Concurrently, fluoroquinolones (e.g., ciprofloxacin, norfloxacin, and levofloxacin) are vital for treating multidrug-resistant (MDR) infections [14]. Fluroquinolone resistance typically arises from chromosomal mutations in *gyrA* and *parC*, which can be vertically transmitted to subsequent bacterial generations [15]. An emerging concern is the co-occurrence of ESBL genes and mutation in *gyr*A and *par*C in *Salmonella* from various sources in different world regions [16], limiting treatment options and leading to a higher risk of treatment failure and complications.

As a key strategy for controlling AMR, quantitative AMR risk assessments (AMR RA) systematically evaluate the probability and impact of AMR, providing insights into exposure risks, infection outcomes, and mitigation effectiveness, to identify high-risk transmission pathways and guide evidence-based interventions and policies [17]. The Codex Alimentarius Commission uses quantitative AMR RA to scientifically assess and manage risks from AMR pathogens in the food chain [17].

Understanding resistance mechanisms, transmission, evolution, and the distinction between intrinsic and acquired resistance is crucial for tracking AMR spread, predicting emerging threats, and designing targeted interventions. Such insights can be gained through genomic analysis using whole genome sequencing (WGS). It is recommended to collaborate AMR genetic data into risk assessment; however, it is not essential for basic calculation [18]. To date, genetic data remains underutilized in AMR RA and should be fully integrated to improve risk assessment. Therefore, this study aimed to quantitatively assess the health effects of consuming pork contaminated with *Salmonella* resistant to third-generation cephalosporins (HGT) and ciprofloxacin (VGT), by integrating complementary genetic data on AMR.

## 2. Results

### 2.1. Salmonella Contamination and Concentration

Overall, *Salmonella* spp. was detected in 42.75% (339/793) of pork samples, with a higher prevalence in samples from fresh markets (75.5%, 305/404) compared to those from hypermarkets (8.74%, 34/389). The *Salmonella* concentration in contaminated pork samples ranged from 1.3 to 180 MPN/g with an average of 18.88 MPN/g and a median of 6.40 MPN/g. Pork from fresh markets contained a higher average concentration (19.59 MPN/g, 1.3–180 MPN/g in range) compared to pork from hypermarkets (13.04 MPN/g, 1.3–140 MPN/g in range).

A total of 531 *Salmonella* isolates were collected, of which 33 serovars were identified. The most common serovars were Rissen (22.6%), followed by Sinstorf (17.8%) and Derby (10.2%).

### 2.2. Antimicrobial Susceptibilities and ESBL Production

Most isolates were resistant to ampicillin (87.95%), florfenicol (56.12%), and streptomycin (54.43%), with lower rates for cefotaxime (31.45%), gentamicin (28.81%), ceftazidime (23.92%), and others (Figure 1). Twelve percent of the isolates were MDR. None were resistant to colistin, imipenem, or meropenem. Eighty-two resistance patterns were observed, of which most common patterns were AMP-OXA (18.46%), AMP-OXA-STR (9.79%), and AMP-FFC-OXA-STR (8.85%).

Twenty-eight percent of all *Salmonella* isolates were confirmed as ESBL producers, accounting for 92.6% of third-generation cephalosporin resistance (*n* = 162). Most (97%) originated from pork in fresh markets, mainly serovar Sintorf (34%). Nearly all were cefotaxime-resistant (99.33%), while 75.33% were ceftazidime-resistant (Figure 1). Up to 32 resistance patterns were defined, of which OXA-FFC-CRO-AMP-FOT-TAZ-GEN-STR (28.66%) was most common. Fluoroquinolone-resistant *Salmonella* were resistant to ampicillin (100%), nalidixic acid (93.55%), streptomycin (93.55%), florfenicol (87.10%), cefotaxime (64.52%), and gentamicin (61.29%) (Figure 1).

### 2.3. Genotype Underlying ESBL Production and Fluoroquinolone Resistance

Among the ESBL producers (*n* = 150), β-lactamase genes in *bla*_TEM_ (66.67%), and *bla*_CTX-M_ (92.67%) groups were identified. Of all *bla*_TEM_-positive isolates (*n* = 100), *bla*_TEM1_ (57%) and *bla*_TEM215_ (43%) were found. Among *bla*_CTX-M_-positive isolates (*n* = 139), *bla*_CTX-M-55_ was most prevalent (84.17%), followed by *bla*_CTX-M-14_ (15.82%).

Point mutations were observed in QRDR of all fluoroquinolone-resistant *Salmonella*, including C-248-T, C-248-A and G-259-A in *gyrA* leading to Ser-83-Phe amino acid substitution (45.16%), Ser-83-Tyr (3.22%), and Asp-87-Asn in GyrA (6.45%), respectively, and G-239-T in *parC* leading to Ser-80-Ile in ParC (6.45%) (Table 1). Isolates with Ser-83-Phe additionally carried *qnrS* (*n* = 2) or both *qnrS* and *aac(6′)-Ib-cr* (*n* = 1). Two isolates simultaneously carried Ser-83-Phe and Asp87Asn in GyrA and Ser80Ile in ParC but lacked PMQR genes. Two others carried Ser83Tyr in GyrA, but of which also harbored *qnrS*. Seven ciprofloxacin-resistant *Salmonella* carried *qnrS* only.

### 2.4. In Vitro Conjugative Transfer of ESBL Genes

Twenty *Salmonella* isolates (two Wandsworth, one Rissen, three Give, and fourteen Sinstorf) yielded transconjugants carrying *bla*_CTX-M-55_, corresponding to their respective donors (Table 2). Donors and transconjugants displayed similar resistance profiles, while minor differences were observed in *Salmonella* Wandsworth (SA149 and SA150) and Sinstorf (SA181, SA219 and SA365) with the absence of azithromycin resistance phenotype in their corresponded transconjugants. The conjugation rate ranged from 3.3 × 10^−6^ to 2.06 × 10^−8^, with an average of 1.06 × 10^−6^.

### 2.5. Genomic and Plasmid Characteristics

#### 2.5.1. Genomic Characteristics of *Salmonella*

Twelve ESBL-producing and fluoroquinolone-resistant *Salmonella* isolates had genome sizes of 5,006,248 bp–5,188,134 bp and GC contents of 51.9–52.2%. The number of genes ranged from 4963 to 5157 (Table A1, Appendix A). The number of protein coding sequences varied from 4627 to 4924. Sequence typing identified five sequence types (STs) including ST198 (Kentucky, *n* = 3), ST13 (Derby and Agona, *n* = 4), ST29 (Stanley, *n* = 1), ST34 (Typhimurium, *n* = 1), and ST1498 (Wandsworth and Mons, *n* = 3) (Table 3).

#### 2.5.2. WGS-Based AMR Phenotype Prediction and Genotypic Analysis

Resistance patterns determined by AST and predicted by WGS (Table 3). Genome analysis revealed several AMR genes, mainly associated with ESBL and fluoroquinolone resistance (Table A2, Appendix B). Each isolate carried 12–18 AMR genes, including at least 1 β-lactamase gene, *bla*_CTX-M-55_ (*n* = 7), *bla*_TEM-1_ (*n* = 9), *bla*_CMY_ (*n* = 3), and *bla*_VEB_ (*n* = 1). The *bla*_CTX-MM_ in SA105 was detected by PCR but not WGS. Mutations in *gyrA* (G259A, C248T, C248A) and *parC* (G239T, C170G) were found. Seven isolates (SA69, SA74, SA81, SA231, SA510, SA523, SA485) carried mutations in both genes, while others had a single QRDR mutation in either *gyrA* or *parC* with *qnrS*1, and SA175 additionally carried *qepA*. Two fluoroquinolone resistance genes were identified in eight isolates, *qnrS* (*n* = 8) *aac(6′)-Ib-cr* (*n* = 3) and *qepA* (*n* = 1), with one isolate carrying both genes (Table 3).

#### 2.5.3. Comparison of R Plasmid

All isolates tested contained at least one plasmid in four incompatibility (Inc) groups including IncHI2/N (*n* = 6), IncC (*n* = 4), IncFII (*n* = 3), and IncN (*n* = 1) (Table 3). Four isolates (SA175, SA485, SA510 and SA523) contained two plasmids including IncC/IncN or IncC/IncFII. The *bla*_CTX-M-55_ gene was carried on IncHI2/N plasmids in serovars Kentucky (SA69, SA74 and SA231), Wandsworth (SA149 and 150), and Mons (SA153) and IncC plasmids in serovars Typhimurium (SA175). IncFII plasmids in SA485, SA510, and SA523 carried no resistance genes. The *qnrS*1 gene was located on IncHI2/N and IncC plasmids in eight isolates, of which SA175 additionally carried *qepA* on IncN plasmid (Table 3).

A comparative analysis of IncHI2/N, IncC, IncN, and IncF plasmids is presented (Figure 2). The IncHI2/N plasmids (pSA69-HI2/N, pSA74-HI2/N, pSA149-HI2/N, pSA150-HI2/N, pSA153-HI2/N, and pSA231-HI2/N) carried two AMR gene clusters, including tet(M)-*bla*_TEM-1_-*bla*_CTX-M-55_-*aac(3)*-IId-*dfrA*12-aadA2-qacE-sul1, flanked by Tn3 and IS26 and *aph(3″)Ib-aph(6)Id-tet(A)-floR-tet*, adjacent to tnpA and insA elements (Figure 2A).

The cluster sequence alignment showed similarity to *E. coli* plasmids from humans in different countries (i.e., pLA065 from Laos, accession number OP242285 and p2017.03.03CC from Vietnam, accession number LC511658) and from pigs (i.e., pD208 from Thailand, accession number LC807803 and p65 from the US, accession number MT077888). The IncHI2/N plasmids exhibited closest similarity to p65 (Figure 3) but differed from pD208 that lacked the tet(M)-KTR-Tn3-*bla*_TEM-1_ region. Both human plasmids (pLA065 and p2017.03.03CC) harbored *bla*_CTX-M-55_ but lacked the *tet*(M)-KTR-Tn3-*bla*_TEM-1_ region as well as class 1 integrons carrying *dfrA*12-aadA gene cassette arrays.

IncC plasmids (i.e., pSA523-C, pSA485-C, pSA81-C, pSA175-C, and pSA510-C) carried int2 together with several AMR genes including *aac(3)-*Via, *ant(3″)-IIa*, *tet(A)*, *floR*, *aph(6)-Id*, *aph(3″)-Ia*, *aph(3″)-Ib*, *bla*_CMY-2_, *bla*_TEM-1_ and *qacE* (Figure 2C). pSA510-C from *S.* Kentucky and pSA523-C and pSA485-C from *S. agona* displayed conserved gene organization whereas alignment gaps were observed in pSA81-C from *S. derby* and pSA175-C in *S. typhimurium*. The *tet(A)*, *floR*, *aph(6)-Id* and *cmy-2* region in the IncC plasmids were similar to those in pSNE3-1928, pSB109, and pFSIS11809860-1.

IncFII plasmids (pSA510-F, pSA523-F, pSA485-F) lacked ESBL genes but encoded virulence genes (virB and hin) (Figure 2D). An IncN plasmid carrying *bla*_TEM-1_ (pSA175-N) was co-harbored with pSA175-C, an IncC plasmid harboring *bla*_CTX-M-55_ (Figure 2B).

#### 2.5.4. Genetic Relatedness of *Salmonella*

Phylogenetic tree of the core genome sequences was constructed for 12 *Salmonella* isolates from pork and deposited into GenBank. The isolates are clustered into five distinct clades: clade 1, SA153; clade 2, SA175; clade 3, SA149; clade 4, SA69, SA231, and SA74; and clade 5, SA105, SA150, SA523, SA81, SA485, and SA510 (Figure 4).

The *Salmonella* isolates in clade 5 clustered with *Salmonella* Kentucky SSSE-138 from a slaughterhouse worker’s hand in Thailand, *Salmonella* Agona R21.2429 from a diarrhea case in Taiwan, and *Salmonella* Agona 8878_Sal_21 from a pig in Thailand.

### 2.6. Exposure Assessment

Of the 793 pork samples, 339 (42.75%) were contaminated with *Salmonella*, including 127 (16%) with ESBL-producing strains and 25 (3%) with ciprofloxacin-resistant strains. Based on the three risk scenarios, the final *Salmonella* concentrations across all cooking conditions followed an exponential distribution. For ESBL-producing *Salmonella* (Scenario 1), the mean final concentrations were 3.34, 2.76 × 10^−2^, 7.57 × 10^−4^, and 6.25 × 10^−6^ CFU/g after 0, 1, 2, and 3 min of cooking, respectively. Similarly, for ciprofloxacin-resistant *Salmonella* (Scenario 2), the mean final concentrations were 4.90 × 10^−1^, 4.05 × 10^−3^, 7.57 × 10^−4^, and 2.76 × 10^−5^ CFU/g. In scenario 3, for all *Salmonella*-contaminated samples, the mean final concentrations were 1.11 × 10^1^, 2.76 × 10^−2^, 7.57 × 10^−4^, and 6.25 × 10^−6^ CFU/g after 0, 1, 2, and 3 min of cooking, respectively.

For the exposure estimation, a daily pork consumption rate of 10.8 g/person/day was assumed, following an exponential distribution. The probability of exposure (Pe) at a 95% confidence interval for cooking times of 0 to 3 min ranged from 9.15 × 10^−1^ to 6.65 × 10^−5^ for ESBL-producing *Salmonella* (scenario 1), 7.11 × 10^−1^ to 3.01 × 10^−4^ for ciprofloxacin-resistant *Salmonella* (scenario 2) and 9.13 × 10^−1^ to 6.41 × 10^−5^ for any *Salmonella* (scenario 3) (Table 4).

### 2.7. Hazard Characterization

The probability of illness (Pi) was estimated at a 95% confidence interval based on the daily dose of ingested bacteria derived from the exposure assessment. A comprehensive statistical analysis was conducted to incorporate the uncertainty associated with the exposure data into the final risk characterization equation.

The average Pi in scenario 1 ranged from 0.05 to 1.77 × 10^−8^, while in scenario 2, it ranged from 0.01 to 7.44 × 10^−7^ across cooking times of 0, 1, 2, and 3 min. In Scenario 3, the probability ranged from 5.10 × 10^−2^ to 1.64 × 10^−7^ over the same cooking durations (Table 4).

### 2.8. Risk Characterization

By combining the probabilities of exposure and illness, the annual risk of illness from consuming pork contaminated with ESBL-producing *Salmonella*, ciprofloxacin-resistant *Salmonella*, and any *Salmonella* was estimated at a 95% confidence interval.

Scenario 1 for ESBL-producing *Salmonella* predicted that the worst-case scenario, consumption of raw pork meat, would result in an average annual risk of 0.8849, equivalent to 88,492 cases per 100,000 population in Bangkok (Table 4). Cooking pork at 64 °C for 1, 2, and 3 min significantly reduced the average annual risk to 9497; 25, and 0 cases per 100,000 population, respectively. In Scenario 2, which focused on ciprofloxacin-resistant *Salmonella* isolates, the predicted average annual illness (Pi) at 64 °C was 63,508; 570; 23; and 0 cases per 100,000 persons, respectively, for cooking times of 0, 1, 2, and 3 min. In Scenario 3, which examined the entire *Salmonella* population, the predicted average annual illness at 64 °C were 95,328; 28,232; 21.51; and 0 cases per 100,000 persons for the same cooking durations.

### 2.9. Scenarios Comparison and Sensitivity

Compared to the uncooked condition, cooking pork for 1 to 3 min significantly reduced the annual risk of illness from ESBL-producing *Salmonella* from 100% to 10.7%, 0.02%, and 0%, respectively, per 100,000 people. Similarly, for ciprofloxacin-resistant *Salmonella*, the risk of illness decreased from 100% to 0.90%, 0.04%, and 0% for the same cooking durations. In scenario 3, the risk of illness from *Salmonella* (QMRA) reduced from 100% to 29.6%, 0.02%, and 0% across the 0, 1, 2, and 3 min of cooking time, respectively, for the same cooking durations. A comparison of the average illness per 100,000 population per year among the three scenarios is shown in Figure 5. ESBL-producing *Salmonella* posed a greater risk than ciprofloxacin-resistant *Salmonella* but a risk comparable to that of all *Salmonella*. Across all scenarios, the risk became negligible after 3 min of cooking.

A sensitivity analysis was conducted to assess the impact of two key input variable factors: initial bacterial concentration and daily pork consumption (Table 5). The R-value represents the magnitude of the impact of each factor on the output variable (risk of illness). In all scenarios, the two factors exhibited similar levels of influence, with most R-values being positive and relatively weak (below 0.5). However, a stronger positive correlation was observed between initial concentration and risk of illness in scenario 1 of ESBL-producing *Salmonella* at condition 2 for 1 min cooking.

## 3. Discussion

A key finding of this study was the high *Salmonella* contamination rate in pork (42.75%) sold in markets. Although this rate was lower than those in previous studies from Thailand and neighboring countries [4,13], it represents a significant public health concern. The prevalence of *Salmonella*-contaminated pork in fresh markets was about nine times higher than in hypermarkets, reflecting comparatively lower hygiene standards in fresh markets. Similar observations have been documented in Thailand [4] and neighboring countries, including Lao PDR, Myanmar [13], Cambodia [19], Vietnam [20], and China [21]. The concentration of *Salmonella* in pork averaged 18.88 MPN/g, in agreement with a previous study in Thailand [22], but this is lower than that of other studies in Cambodia, Vietnam, and China [19,21,23]. This quantitative data is crucial for microbial risk assessment, as higher concentrations increase the risk of human illness. These findings underscore the need to further strengthen food safety practices and sanitation measures at points of sale.

While serovar Rissen has been widely reported in pork and poultry across Europe and Southeast Asia [24,25,26], serovar Sinstorf emerged as the second-most common serovar in this study (16.57%). Historically, serovar Sinstorf was rarely reported but its prevalence has been increasingly reported [27], suggesting a potential shift in serovar distribution. This serovar grows rapidly but exhibits reduced pathogenicity due to the lower expression of its virulence genes [28]; however, factors behind its raising prevalence remain unclear. The serovar Sinstorf exhibited MDR phenotype, with over half (51/89) producing ESBLs and capable of transferring ESBL genes. In contrast, only 4% of the most common serovar, Rissen, were ESBL producers, similar to a previous study [27]. However, the phenotypic and genotypic AMR profiles were not statistically associated with serovar Sinstorf. Further studies are warranted to elucidate the underlying characteristics of this serovar.

The prevalence of ESBL producers (28.2%) in this study was comparable to a previous report in pork from Cambodia [29], higher than those in Thailand [4,13] but lower than in China [30] and Portugal [31]. Concurrently, ciprofloxacin resistance rate in this study was rather low (5.8%), above a previous report [4] but below that in fresh markets in Northern Thailand [32], China, Brazil, and the USA [33,34,35].

This study conducted a quantitative AMR risk assessment using a market-to-fork model based on bacterial concentrations in pork and consumption patterns. Genetic AMR data for a subset of isolates was in cooperation to complement hazard identification and exposure assessment.

HGT is expected to pose a higher AMR risk by promoting broader dissemination and the emergence of new resistant strains, whereas VGT maintains AMR within populations over time [36]. Consistently, illness from consuming raw pork meat contaminated with ESBL-producing *Salmonella* (88,194 per 100,000 population per year) exceeded ciprofloxacin-resistant *Salmonella* (61,877 per 100,000 population per year), supported by the conjugal transfer of *bla*CTX-M-55 carrying plasmids. These results highlight the importance of antimicrobial stewardship in controlling AMR propagation along the food chain.

The QMRA was extended to assess the risk of *Salmonella* in pork, allowing comparison of the potential health impacts of generic versus AMR strains. The risk of illness from consuming raw pork meat contaminated with ESBL-producing *Salmonella* was comparable to that from raw pork meat contaminated with any *Salmonella* (95,328.26 per 100,000 population per year). In contrast, the lowest risk of illness was observed for consuming pork contaminated with ciprofloxacin-resistant *Salmonella*, primarily due to its low prevalence and limited clonal spread.

Due to the unavailability of mortality data for infections caused by ESBL-producing and ciprofloxacin-resistant *Salmonella*, the risk of illness was used for the assessment. *Salmonella* infections rarely cause death, except in vulnerable populations such as young children, the elderly, and people with complications or compromised immunity. However, AMR can increase the risk of death both directly and indirectly [37]. Moreover, the mortality data for Salmonellosis from studies conducted over two decades ago and from different geographical regions may not accurately reflect the current trends. The use of such data could lead to overestimation or underestimation of the actual risk.

Cooking pork at 64 °C for one minute significantly reduced the risk, and for 3 min eliminated the risk of illness in all scenarios, in agreement with a previous study [38]. The fixed time intervals of 0, 1, 2, and 3 min were used to estimate bacterial reduction, rather than variable durations to ensure result clarity and facilitate effective risk communication with the public. While there was no report of AMR RA of ESBL-producing or cephalosporin-resistant *Salmonella* in Thailand, a previous study of ciprofloxacin-resistant *Salmonella* from retail pork was conducted in Northern Thailand [38]. The latter showed that cooking above 60 °C effectively reduced bacterial counts in pork products and 2 min cooking at 64 °C eliminated all risk.

All ESBL-producing *Salmonella* in this study were MDR, consistent with a previous study [12] and likely reflecting co-localization of resistance genes and efficient horizontal gene transfer via plasmids. Almost all ESBL producers carried either *bla*_CTX-M-55_ or *bla*_CTX-M-14_, indicating a strong concordance between phenotypic and genotypic resistance. This gene has been frequently reported in *Salmonella* from pork and humans in multiple countries, suggesting its widespread dissemination through the food chain [29,39].

Under ampicillin selective pressure, *bla*_CTX-M-55_ in all ESBL-producing *Salmonella* donors were found to be located on conjugative plasmids, consistent with previous observations in both meat-derived and clinical human isolates [40]. All transconjugants exhibited MDR phenotype, indicating the co-localization of resistance genes on the same plasmid. These findings suggest that *E. coli* from pork can serve as a reservoir of ESBL genes capable of transferring to other bacteria, highlighting the public health risk associated with transmission through meat consumption. These findings confirm that *Salmonella* and its resistance determinants are hazards for risk assessment [17]. The co-selection of AMR genes by a single antibiotic further emphasizes that sustainable control of AMR demands a coordinated, whole system approach to the regulation of antimicrobial use.

WGS of the selected isolates provided comprehensive information on the AMR characteristics of each isolate, allowing comparative analysis with genomes or plasmid sequences from humana and pig sources. In addition to supporting hazard identification, such comprehensive genomic information strengthens AMR risk assessment by identifying potential reservoirs and transmission routes, improving the understanding of exposure pathways, and supporting evidence-based evaluation of the likelihood and consequences of AMR dissemination through the food chain. Comparative analysis of plasmids revealed that shared backbones and resistance gene arrays among IncHI2/N and IncC plasmids reflect conserved core structures, indicating that their spread is driven by horizontal gene transfer. Despite their high similarity, IncHI2/N plasmids in this study carried the AMR gene clusters *aph(3″)Ib-tet(A)-floR-aph(6)Id* that are absent in p65 and may reflect differences in antimicrobial use across geographic regions. Co-occurring plasmids in different Inc groups (IncN, IncC and IncFII[pMET]) in the same isolates (i.e., SA175, SA485, SA510, and SA523) may enhance the dissemination and stability of AMR genes through co-selection and horizontal transfer, supporting the intersectoral dissemination of MDR *Salmonella* by promoting bacterial fitness, adaptability, and persistence and posing public health concerns.

Fluoroquinolone-resistant *Salmonella* in this study carried GyrA substitutions Ser83Phe/Tyr and Asp87Asn, consistent with a previous report on pigs [32] and patients in Thailand [41]. Further investigations are required to determine whether these findings reflect the persistence and possible transmission of resistant clones among meat, animals, and humans in the country. The ParC substitution Ser80Ile was detected for the first time in Thailand but has been previously reported in animal and human isolates elsewhere [42,43]. Although combined GyrA–ParC mutations are known to confer high-level resistance, it was not the case for this study [44]. The frequent finding of *qnrS* (58%) agrees with previous findings in pork, beef, and chicken [11,45,46]. PMQR genes alone usually confer low-level fluoroquinolone resistance but can promote the selection of *gyrA* and *parC* mutations [47], and co-select other AMR genes [48,49].

The phylogenomic analysis demonstrated possible clonal transmission of *Salmonella* among pigs, pork, and humans. The close genetic relatedness between the *Salmonella* isolates from pork in this study and those from a human diarrhea clinical case provides evidence of zoonotic transmission, despite originating from different countries. Importantly, the isolates were clustered with those from the hand of a slaughterhouse worker and a pig in Thailand, indicating the potential cross-contamination during the slaughtering process and persistence of resistant strains along the pork production chain. This finding highlights the foodborne risks associated with AMR *Salmonella* and underscores the need for strengthened hygiene practices and integrated surveillance from farm to slaughterhouse to reduce transmission within the pork production system.

The sensitivity analysis indicated that bacterial concentration and pork consumption quantities had moderate-positive effects on the risk of illness, inconsistent with a previous study highlighting the highest risk factor derived from cross-contamination such as wash meat before cooking or sharing utensils [19].

This risk model used in this study offers flexibility for application to other foodborne bacteria and in other geographical contexts, with appropriate parameter adjustments. Limitations of this study include the challenges of model validation due to limited data on these specific resistant *Salmonella* strains, reliance on some outdated data sources, the absence of uncertainty and variability analyses, and the use of in vitro conjugation data, which may not fully reflect in vivo transfer dynamics. Therefore, further studies are warranted to strengthen the validation of this risk model and better assess its predictive accuracy. The validation could be improved by integrating longitudinal surveillance of *Salmonella* and AMR along pork supply chains, using more realistic data on horizontal gene transfer in animals, and incorporating variability and uncertainty analyses through probabilistic simulations. Comparing the model with independent datasets and human salmonellosis data would provide a better assessment of its performance, while testing it against field-based intervention studies would further support its practical applicability.

Several countries have culinary traditions that include the consumption of raw or undercooked pork, e.g., Thailand, Japan, China, Vietnam [50,51]. A recent survey revealed that 66% of the Thai population regularly consumes undercooked meat, with nearly 80% of these individuals residing in the northeastern and northern regions of the country [52]. This dietary habit heightens the risk of exposure to foodborne pathogens. Therefore, targeted risk communication and culturally sensitive interventions are crucial to encourage safe cooking practices.

In conclusion, consumption of raw pork meat poses the highest risk, reinforcing the importance of thoroughly cooking meat. Communication with consumers, focusing on safe food handling, and thorough cooking is crucial to mitigate the risks associated with *Salmonella*-contaminated pork. Public education and awareness regarding the risks of eating raw pork meat should be promoted. The data could beneficially inform policymakers for risk management. While interventions at the point of sale are important, a comprehensive approach encompassing the entire pork production chain, from farm to fork, is essential for effective AMR control.

## 4. Materials and Methods

### 4.1. Sampling Design and Sample Collection

A total of 793 pork samples were collected during 2021–2022 from fresh markets (*n* = 404) and hypermarkets (*n* = 389) in Bangkok, exceeding the calculated sample size of 770 based on an estimated *Salmonella* prevalence of 50%, with a 95% confidence interval, and a 5% margin of error. This cross-sectional study utilized a stratified sampling design, with districts as the stratification units. Nineteen of Bangkok’s 50 districts were proportionally selected based on population density to ensure representative geographic coverage, encompassing at least 60% of the city’s population [53,54]. Each district has 1 retail market and 1–2 hypermarkets. Eighty-three vendors from 43 participating markets were sampled, with each vendor being visited once.

Pork samples (~300 g each) from retailed markets and a whole pack from hypermarkets were aseptically collected at the point of sale. Each was individually placed in a sterile, sealed double-layered plastic bag and stored on ice. The samples were delivered to the laboratory within four hours and analyzed immediately upon arrival.

### 4.2. Salmonella Isolation and Confirmation

*Salmonella* was isolated and biochemically confirmed, following ISO 6579-1:2017 [55]. Three to five *Salmonella* colonies from each positive pork sample were serotyped by slide agglutination [56] using commercial antisera (S & A Reagents Lab, Bangkok, Thailand). A single colony of each serovar from each positive sample was stored in 20% glycerol solution at −80 °C for future analysis.

### 4.3. Salmonella Enumeration

*Salmonella* enumeration was performed in all pork samples using the miniaturized Most Probable Number (MPN) method [57]. Briefly, a 25 g sample was homogenized in 225 mL of Buffered Peptone Water (BPW) to achieve a 1:10 dilution. The sample was serially diluted in a 12-well titer plate with triplicate dilutions and incubated at 37 °C for 18 h. The wells were examined for the presence of *Salmonella* as described above. The number of positive wells across four plate columns was used to determine MPN values using Jarvis’s software version 6 [58] as recommended by ISO7218:2024 [59]. MPN values were converted to CFU using a correction factor of 0.8 to align with dose–response model calculations [60].

### 4.4. Determination of Antimicrobial Susceptibilities and ESBL Production

A total of 531 *Salmonella* isolates were obtained in this study and tested for antimicrobial susceptibilities using the automated Sensititre™ system (Thermo Fisher Scientific, Cleveland, OH, USA) with Asia surveillance plates ASSECAF (TREK Diagnostic Systems, West Sussex, UK). The antimicrobial panel (abbreviation, clinical breakpoint in parenthesis) included cefotaxime (FOT, 4 µg/mL), ceftazidime (TAZ, 16 µg/mL), ciprofloxacin (CIP, 4 µg/mL), nalidixic acid (NAL, 32 µg/mL), gentamicin (GEN, 16 µg/mL), streptomycin (STR, 16 µg/mL), meropenem (MERO, 4 µg/mL), azithromycin (AZI, 16 µg/mL), ampicillin (AMP, 32 µg/mL), and colistin (COL, 2 µg/mL). Disk diffusion method was conducted for cefoxitin (30 µg), cefepime (30 µg), ceftriaxone (30 µg), levofloxacin (5 µg), florfenicol (30 µg), imipenem (10 µg), and oxacillin (1 µg) [60]. All antimicrobial disks were commercially obtained (BD Difco^TM,^ Becton, Dickinson and Company, Sparks, MD, USA). Interpretive criteria were according to CLSI [61]. *Escherichia coli* ATCC 25922, *Pseudomonas aeruginosa* ATCC 27853, and *Staphylococcus aureus* ATCC 29213 served as quality control strains. Multidrug resistance (MDR) is defined as being resistant to at least three antimicrobials from different classes.

The *Salmonella* isolates resistant to ceftazidime and/or cefotaxime were subsequently confirmed for ESBL production using the combination disk method. The difference in ≥5 mm in the inhibition zone between cephalosporin/clavulanic acid combination disks and cephalosporin disks alone was interpreted as ESBL-positive. *Klebsiella pneumoniae* ATCC 700603 was used as a quality control strain.

### 4.5. PCR and DNA Sequencing

Template DNA was prepared using whole-cell boiled lysates [62]. Primers used in this study are listed in Table 6. The PCR reaction for QRDR genes consisted of pre-denaturation at 94 °C for 5 min, followed by 30 cycles of denaturation at 94 °C for 45 s, annealing at 57 °C for 45 s, and extension at 72 °C for 1 min, with final extension at 72 °C for 5 min. Identical conditions were used for *qnrA*, *qnrB*, and *qnrS* genes except *aac(6′)-Ib-cr* (annealing at 55 °C for 45 s) and *qepA* (denaturation at 95 °C for 1 min, annealing at 60 °C for 1 min). For *bla*_CTX-M_, the protocol consisted of pre-denaturation at 94 °C for 3 min, 25 cycles of denaturation at 94 °C for 1 min, annealing at 60 °C for 1 min, extension at 72 °C for 1 min, and final extension at 72 °C for 10 min. Subgroups 1, 2, and 8/25 required 5 min pre-denaturation and annealing at 56 °C, while subgroup 9 used 5 min pre-denaturation with annealing at 60 °C. *bla*_TEM_ and *bla*_SHV_ followed the same conditions as *bla*_CTX-M_ but with annealing at 50 °C. PCR amplicons were purified using NucleoSpin^®^ Gel and PCR Clean-up Kit (Macherey-Nagel, Düren, Germany) and submitted for nucleotide sequencing at Bionics laboratory (Seoul, Republic of Korea). The DNA sequences were analyzed using BLAST version 2.15.0 searches against the GenBank database available at the National Center for Biotechnology Information (NCBI) website.

#### 4.5.1. Detection of β-Lactamase Genes

All ESBL-producing *Salmonella* isolates (*n* = 150) were screened for three *bla* gene groups including *bla*_CTX-M_, *bla*_TEM_, and *bla*_SHV_ using specific primers [67,68]. The CTX-M subgroups were further determined using primers specific for groups 1, 2, 9, and 8/25 [69]. The PCR products were purified and subjected to nucleotide sequencing.

#### 4.5.2. Detection of Quinolone Resistance Mechanisms

All fluoroquinolone-resistant *Salmonella* isolates (*n* = 31) comprising those resistant to ciprofloxacin only (*n* = 15), levofloxacin only (*n* = 3), and both ciprofloxacin and levofloxacin (*n* = 13) were examined for mutations in *gyrA* and *parC* [63]. Two fluoroquinolone-susceptible isolates were included as controls. The nucleotide sequences were compared to the published *gyr*A (GenBank accession number 1253794) and *par*C (GenBank accession number 1254697) of *Salmonella* Typhimurium str. LT2 NC_003197.2. The PMQR genes, i.e., *qnrA*, *qnrS*, *aac(6′)-Ib-cr*, and *qepA*, were PCR-screened in all the isolates (*n* = 531) [64,65,66]. Previously confirmed *Salmonella* isolates carrying the PMQR genes were used as positive controls [71].

### 4.6. Conjugation Transfer of ESBL Genes

Plasmid transfer was evaluated by biparental filter mating following standard protocols. The MDR ESBL-producing *Salmonella* positive (*n* = 30) served as donors. The rifampicin-resistant *E. coli* K12 strain MG1655 (MG1655rif^r^; MIC = 256 µg/mL) served as the recipient. Transconjugants were selected on LB agar containing rifampicin (32 μg/mL) and ampicillin (100 μg/mL). The presence of ESBL genes in transconjugants was confirmed by PCR.

### 4.7. Whole Genome Sequencing (WGS) and Bioinformatics

#### 4.7.1. DNA Extraction

Genomic DNA was extracted from ESBL-producing and fluoroquinolone-resistant *Salmonella* isolates with MDR phenotype (*n* = 11) and one non-ESBL-producing, fluoroquinolone-resistant isolate with MDR characteristics using ZymoBIOMICS DNA kits (Zymo Research Corp., Irvine, CA, USA) (Table 3).

#### 4.7.2. Whole Genome Sequencing

The integrity of the extracted genomic DNA was evaluated by electrophoresis. DNA quality and concentration were determined using a NanoDrop™ 1000 spectrophotometer (Thermo Scientific, Wilmington, DE, USA). WGS was performed using long-read sequencing with Oxford Nanopore Technologies (ONT) at the Siriraj Long-read Lab, Faculty of Medicine Siriraj Hospital, Mahidol University, Bangkok, Thailand, complemented by short-read sequencing with the Illumina HiSeq platform at GENEWIZ, Suzhou, China. Read quality was assessed using NanoPlot [72] and FastQC [73]. Hybrid genome assembly was performed with Unicycler [74], and annotation was carried out using the NCBI Prokaryotic Genome Annotation Pipeline [75]. The annotated genomes were deposited under BioProject accession PRJNA946550.

#### 4.7.3. Genomic Analysis

AMR genes were identified using Resfinder [76] and point mutations in quinolone resistance-determining regions (QRDRs) were detected with PointFinder [77]. Plasmid replicon types were determined using PlasmidFinder [78]. Sequence types and serovars were confirmed by MLST and SeqSero, respectively [40]. Comparative plasmid analysis was performed with Proksee [79] and Clinker [80] to align assembled plasmids from pork isolates, with reference plasmids retrieved from GenBank, including p65 (CP037991), pLA065 (AP018139), and p2017.03.03CC (LC483178) for IncHI2/N and pHNTS45-1 (CP070282) and pN268-2 (CP031658) for IncC.

Phylogenetic analysis was conducted by integrating publicly available *Salmonella* isolates from humans (*S.* Agona R21.2429, accession no. SAMN24782759, Taiwan and *S.* Kentucky SSSE-136, accession no. SAMN08386787, Bangkok, Thailand) and pigs (*S*. Agona 8878_Sal21, accession no. SAMN14945123, Bangkok, Thailand). A maximum-likelihood tree based on core genome SNPs was generated using CSI Phylogeny [81] and visualized with iTOL v6 [82].

### 4.8. Risk Assessment Models

#### 4.8.1. Model Overview

The health risks of infection and illness from consuming pork contaminated with third-generation cephalosporins and fluoroquinolone-resistant *Salmonella* were assessed following the Codex Alimentarius guidelines CXG 77-2011 [17]. ESBL-producing and ciprofloxacin-resistant *Salmonella* represented resistance to third-generation cephalosporins and fluoroquinolones, respectively. Three risk scenarios were established: Scenario 1, ESBL-producing *Salmonella* representing HGT of AMR; Scenario 2, ciprofloxacin-resistant *Salmonella* representing VGT; and Scenario 3, for the quantitative microbial risk assessment of *Salmonella* from pork (Figure 6). Scenario 3 served as a baseline for comparing the AMR risk estimates. The outcomes representing the annual risk of infection per person were modeled with an exponential distribution and analyzed using @Risk v8.1 (Palisade Corporation, Ithacha, NY, USA). All equations (Eq.) used in this study are listed in Table 7.

#### 4.8.2. Exposure Assessment

Initial concentration variable

The initial contamination level or concentration (Cc, CFU/g) was determined by enumerating *Salmonella* in all 793 pork samples. The data was analyzed using @Risk program. The exponential (Expon) distribution model was identified as the best fit and applied to estimate the concentration variables.

2.Cooking reduction model

Raw pork meat is typically cooked before consumption; however, certain traditional Thai dishes involve eating pork raw or only partially cooked. Proper cooking can effectively destroy pathogens such as *Salmonella*. A Log Reduction (LR) approach was applied at a cooking temperature of 64 °C, in line with FAO guidelines [83] (Equation (1)).

3.Remaining concentration variable

The remaining concentration of *Salmonella* in pork (RCc) after cooking at 64 °C for 0, 1, 2, and 3 min was calculated by multiplying the LR with the initial concentration variable (Equation (2)).

4.Consumption variable

The consumption variable (Cs) represents the daily pork consumption by Thai individuals obtained from the Agricultural Commodity and Food Standard in 2016. The variability in Cs was modeled using an exponential distribution. Subsequently, the dose of infection (D, CFU per day), that is, the daily dose of *Salmonella* ingested by an individual in Bangkok, was calculated (Equation (3)).

5.Probability of exposure

The exposure assessment was to determine the probability of an individual consuming at least one CFU of *Salmonella*, either ESBLs-producing, ciprofloxacin-resistant, or any *Salmonella* from pork. The probability of exposure (Pe) was calculated using the daily ingestion dose of *Salmonella* (*D*) (Equation (4)).

#### 4.8.3. Hazard Characterization

Hazard was defined as the probability of illness (Pi) due to the consumption of pork contaminated with *Salmonella*, either ESBLs-producing, ciprofloxacin-resistant, or any *Salmonella* from pork. The dose–response model was employed to define Pi [84] (Equation (5)).

#### 4.8.4. Risk Characterization

The results from the exposure assessment (*Pe*) and hazard characterization (*Pi*) were used to estimate the daily probability of illness (*Piday*) (Equation (6)), assuming that exposure and illness occur independently. Since health data is typically reported in annual units, the annual probability of illness (*Piyear*) was estimated by using a binomial distribution. The *Piyear* simulations were performed 10 times with 10,000 iterations, using Monte Carlo methods for risk prediction (Equation (7)).

#### 4.8.5. Evaluation of Four Cooking Conditions

The three risk scenarios representing the consumption of pork contaminated with ESBL-producing, ciprofloxacin-resistant *Salmonella* or all *Salmonella* were assessed under four different cooking times at 64 °C: 0 min (uncooked), 1 min, 2 min, and 3 min.

#### 4.8.6. Sensitivity Analysis

The impact of each input variable across different conditions in each scenario was evaluated and factors with high impact on human illness were identified. Spearman rank correlation coefficients were calculated using Monte Carlo simulations with 10,000 iterations.

## Figures and Tables

**Figure 1 antibiotics-14-01198-f001:**
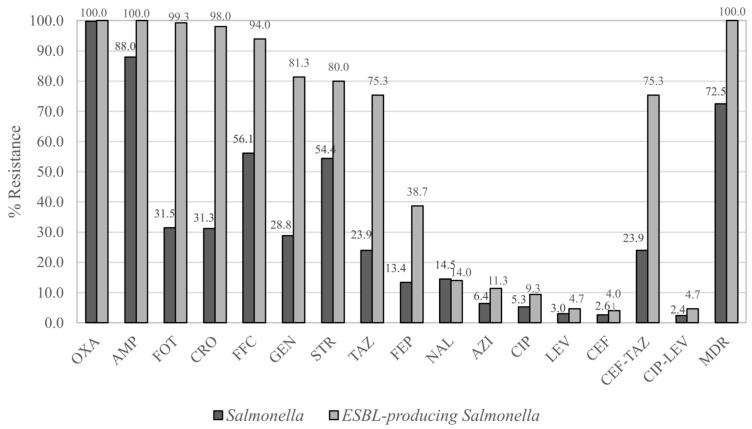
Resistance rates of *Salmonella* and ESBL-producing *Salmonella*. Each antibiotic (abbreviation in three letters) shows resistance rate in percentage among *Salmonella* (
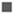
) (*n* = 531) and ESBL-producing *Salmonella* (
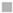
) (*n* = 150). CEF-TAZ stands for resistance to both cefotaxime and ceftazidime as screening of ESBL-producing. MDR stands for resistance to more than three groups of antibiotics.

**Figure 2 antibiotics-14-01198-f002:**
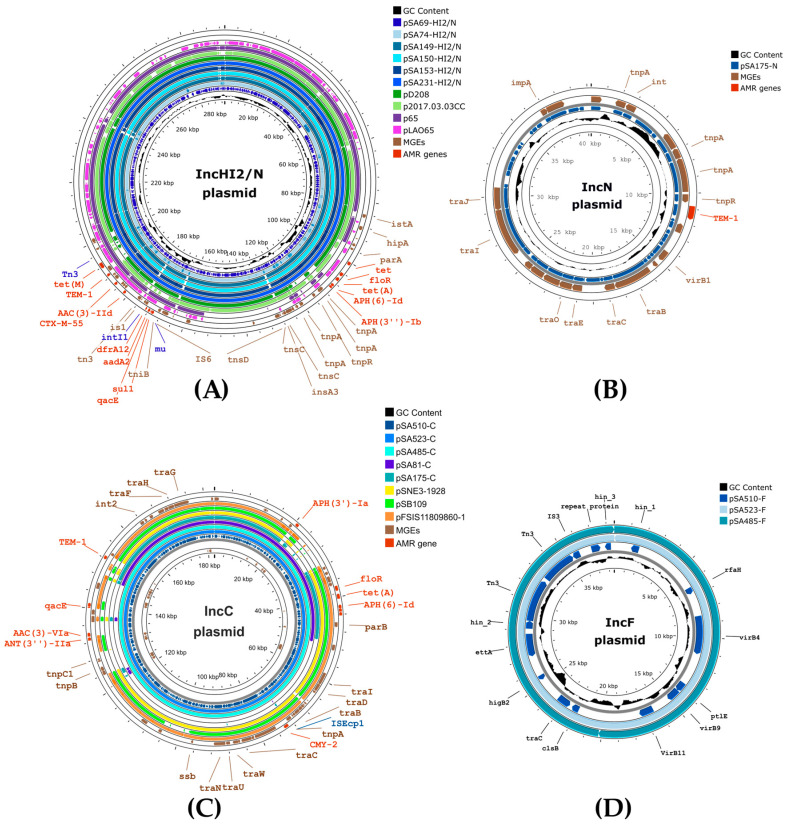
Circular alignment of plasmid replicons. (**A**) Comparison of IncHI2/N plasmids (pSA69-HI2/N, pSA74-HI2/N, pSA149-HI2/N, pSA150-HI2/N, pSA153-HI2/N, and pSA231-HI2/N) from this study with reference plasmids p65 (pig, USA), pD208 (human, Thailand), p2017_03_03CC (human, Thailand), and pLAO65 (human, Lao PDR) revealed two AMR clusters including “*tet(M)*-*bla*_TEM-1_*-bla*_CTX-M-55_-*aac(3)-IId-dfrA*12-*aadA*2-*qacE*-*sul1*” and “*aph(3″)Ib-aph(6)Id-tet(A)*-*floR*-*tet*”. The first region, containing *bla*_TEM-1_ and *bla*_CTX-M-55_, was conserved across all plasmids, while the second region showed variation, particularly in p65. (**B**) IncN plasmid pSA175-N carried *bla*_TEM-1_. (**C**) Comparison of IncC plasmids (pSA510-C, pSA523-C, pSA485-C, pSA81-C, and pSA175-C) from this study with plasmids from different geographic regions and hosts including pSNE3-1928 (human, USA), pSB109 (pig, Spain), and pFSIS11809860-1 (pig, USA) revealed that the *tet(A)*, *floR* and *aph(6)-Id* gene cluster was present in all plasmids. (**D**) IncF plasmids pSA510-F, pSA523-F, and pSA485-F from this study carried virulence-associated genes (*vir*).

**Figure 3 antibiotics-14-01198-f003:**
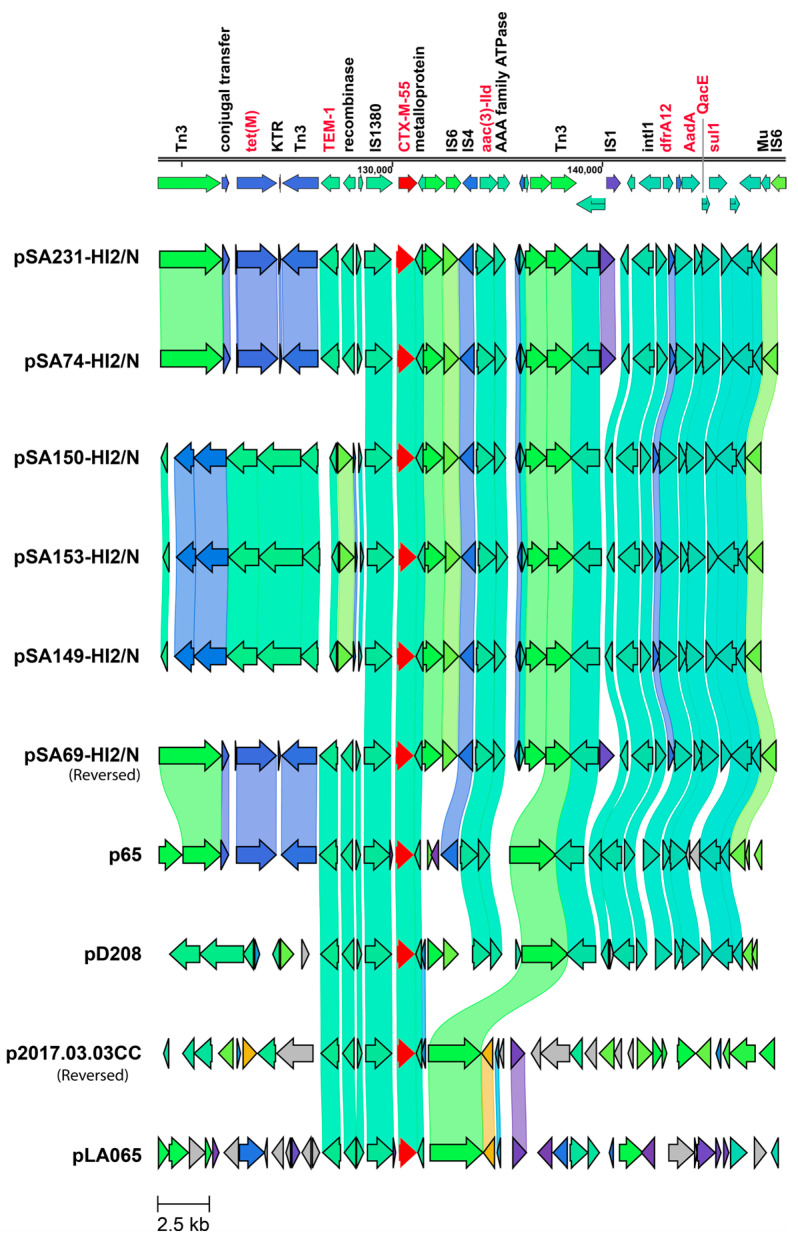
Genomic comparison of IncHI2/N plasmids highlighting conserved MDR region containing *bla*_CTX-M-55._ Homologous sequences are shown and linked in the same color. Sequences from IncHI2/N plasmid pSA69-HI2/N, pSA74-HI2/N, pSA149-HI2/N, pSA150-HI2/N, pSA153-HI2/N, and pSA231-HI2/N in this study were aligned with reference plasmids from p65, pD208, p2017_03_03CC, and pLAO65. Genes highlighted in red font represent AMR genes. The ESBL gene *bla*_CTX-M-55_ is indicated by red arrow. The direction of the arrow indicates the direction of gene transcription.

**Figure 4 antibiotics-14-01198-f004:**
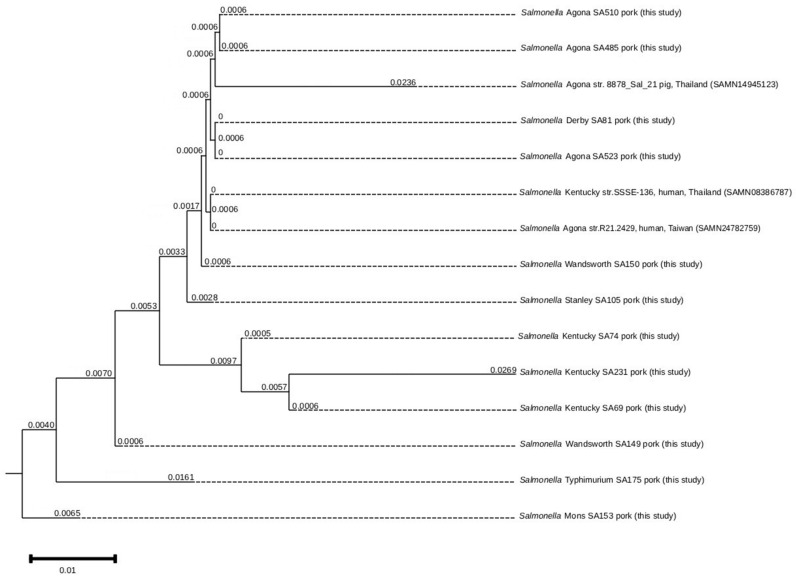
Phylogenetic tree by WGS analysis among *Salmonella* isolates (*n* = 18). Chromosomal sequences of twelve *Salmonella* SA69, SA74, SA81, SA105, SA149, SA150, SA153, SA175, SA231, SA485, SA510, and SA523 from pork in this study were aligned with *S. Agona* R21.2429 (human, Taiwan) *S. Kentucky* SSSE-136 (human, Thailand) and *S. Agona* 8878_Sal21 (pig, Thailand). The isolates clustered into five distinct clades: clade 1, SA153; clade 2, SA175; clade 3, SA149; clade 4, SA69, SA231, and SA74; and clade 5, SA105, SA150, SA523, SA81, SA485, and SA510. The numerical values displayed on the branches represent branch lengths, scaled to the number of substitutions per site.

**Figure 5 antibiotics-14-01198-f005:**
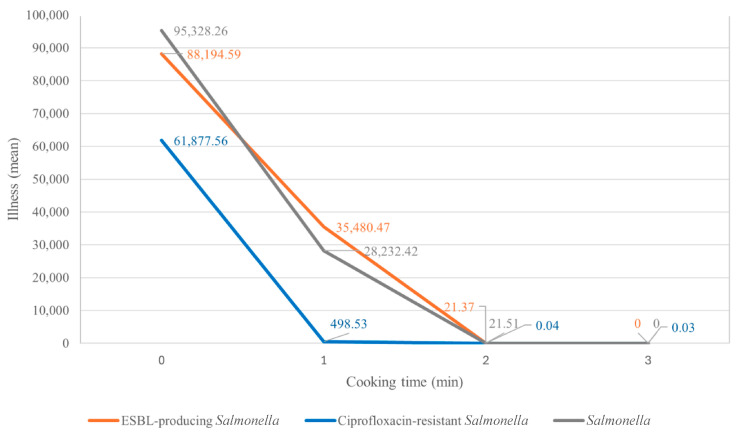
The probability of illness from consuming pork contaminated with *Salmonella* per 100,000 population per year. The lines show trends of probability of illness (mean) sorted by cooking time (0, 1, 2 and 3 min) including three scenarios: antimicrobial-resistant risk assessment of ESBL-producing *Salmonella* (orange line), ciprofloxacin-resistant *Salmonella* (blue line), and QMRA *Salmonella* (gray line).

**Figure 6 antibiotics-14-01198-f006:**
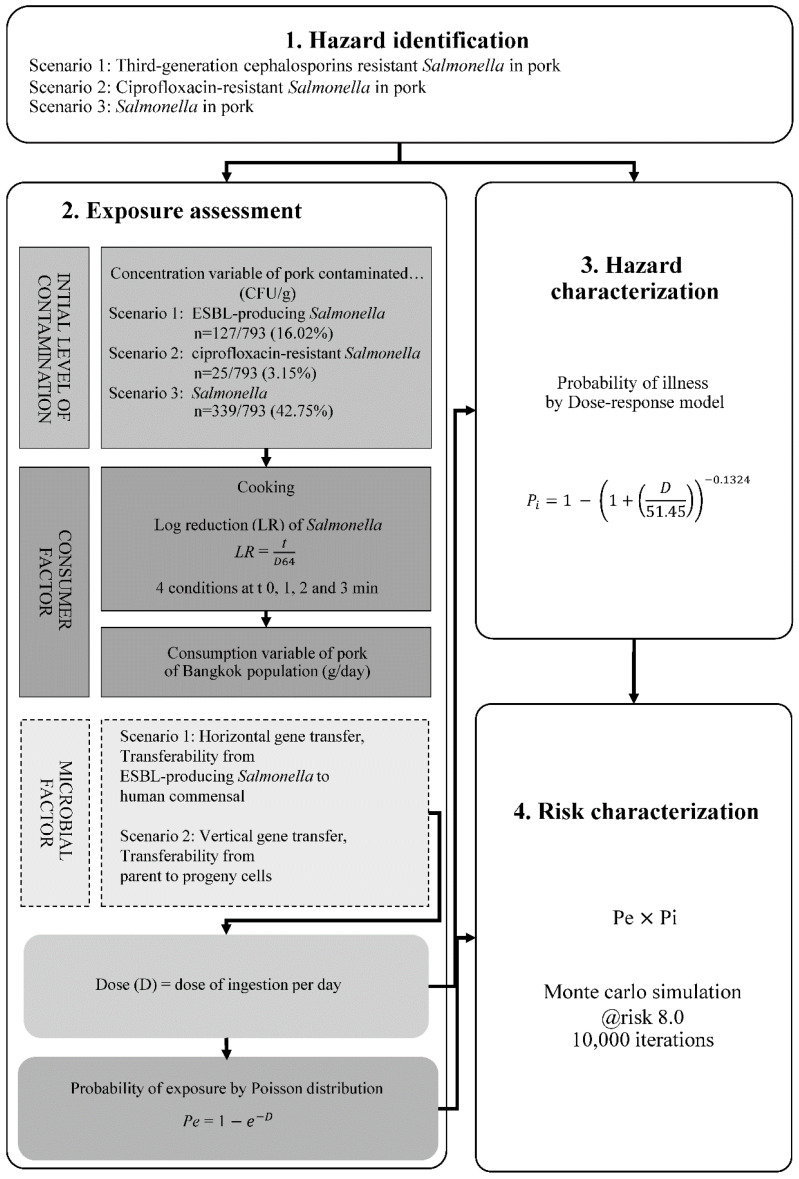
Risk assessment (RA) model of *Salmonella*-contaminated pork (*n* = 793) in 3 scenarios. Scenario 1: RA for ESBL-producing *Salmonella* (*n* = 127), Scenario 2: RA for ciprofloxacin-resistant *Salmonella* (*n* = 25), Scenario 3: RA for *Salmonella* (*n* = 339).

**Table 1 antibiotics-14-01198-t001:** Mutations in *gyrA* and *parC* of fluoroquinolone-resistant *Salmonella* (*n* = 31).

ID	QRDR ^a^ Mutation in Gene (Amino Acid Change)		PMQR ^a^ Gene	Resistance Pattern (*n*)
	*gyrA*(GyrA)	*parC*(ParC)		
1	C248T(Ser83Phe)	ND ^a^	*qnrS*	CIP ^a,b^-LEV (1)
			*qnrS-aac(6′)-Ib-cr*	CIP ^a,b^-LEV (2)
			ND	CIP (6) ^c^, CIP ^c^-LEV (5)
2	C248T(Ser83Phe)	G293T(Ser80Ile)	ND	CIP ^a^-LEV (2)
	G259A(Asp87Asn)			
3	C248A(Ser83Tyr)	ND	*qnrS*	CIP ^b^-LEV (1) LEV (1)
4	ND	ND	*qnrS*	CIP ^b^ (7)
			ND	CIP ^b^ (2), CIP ^c^-LEV (2)

^a^ abbreviation: CIP, ciprofloxacin; LEV, levofloxacin; PMQR, Plasmid-Mediated Quinolone Resistance; QRDR, Quinolone Resistance-Determining Region. ND, not detected. ^b^ CIP MIC, 16 µg/mL. ^c^ CIP MIC, 4–16 µg/mL.

**Table 2 antibiotics-14-01198-t002:** Characteristics of ESBL-producing *Salmonella* donors and transconjugants in conjugation experiments (*n* = 20).

ID.	Serovars	Donors		Transconjugants	
		Resistance Pattern ^a^	ESBL Genes	Resistance Pattern	ESBL Genes
SA149	Wandsworth	AMP-AZI-CIP-CRO-FEP-FFC-FOT-GEN-NAL-OXA-STR-TAZ	*bla* _CTX-M-55_	AMP-AZI-CRO-FEP-FFC-FOT-GEN-OXA-STR-TAZ ^b^	*bla* _CTX-M-55_
SA150	Wandsworth	AMP-AZI-CIP-CRO-FEP-FFC-FOT-GEN-NAL-OXA-STR-TAZ	*bla* _CTX-M-55_	AMP-AZI-CRO-FEP-FFC-FOT-GEN-OXA-STR ^b^	*bla* _CTX-M-55_
SA24	Rissen	AMP-CRO-FOT-GEN-OXA-STR-TAZ	*bla* _CTX-M-55_	AMP-CRO-FOT-GEN-OXA-STR-TAZ	*bla* _CTX-M-55_
SA75	Give	AMP-CRO-FEP-FFC-FOT-GEN-OXA-STR-TAZ	*bla* _CTX-M-55_	AMP-CRO-FEP-FFC-FOT-GEN-OXA-STR-TAZ	*bla* _CTX-M-55_
SA82	Give	AMP-CRO-FFC-FOT-GEN-OXA-STR-TAZ	*bla* _CTX-M-55_	AMP-CRO-FFC-FOT-GEN-OXA-STR-TAZ	*bla* _CTX-M-55_
SA90	Sinstorf	AMP-CRO-FFC-FOT-GEN-OXA-STR-TAZ	*bla* _CTX-M-55_	AMP-CRO-FFC-FOT-GEN-OXA-STR-TAZ	*bla* _CTX-M-55_
SA113	Sinstorf	AMP-CRO-FFC-FOT-GEN-OXA-STR-TAZ	*bla* _CTX-M-55_	AMP-CRO-FFC-FOT-GEN-OXA-STR-TAZ	*bla* _CTX-M-55_
SA173	Sinstorf	AMP-CRO-FEP-FFC-FOT-GEN-OXA-STR-TAZ	*bla* _CTX-M-55_	AMP-CRO-FEP-FFC-FOT-GEN-OXA-STR-TAZ	*bla* _CTX-M-55_
SA181	Sinstorf	AMP-CRO-FFC-FOT-GEN-OXA-STR-TAZ	*bla* _CTX-M-55_	AMP-AZI-CRO-FFC-FOT-GEN-OXA-STR ^b^	*bla* _CTX-M-55_
SA202	Sinstorf	AMP-CRO-FFC-FOT-OXA-STR-TAZ	*bla* _CTX-M-55_	AMP-CRO-FFC-FOT-OXA-STR-TAZ	*bla* _CTX-M-55_
SA219	Sinstorf	AMP-CRO-FEP-FFC-FOT-GEN-OXA-STR-TAZ	*bla* _CTX-M-55_	AMP-AZI-CRO-FEP-FFC-FOT-GEN-OXA-STR-TAZ ^b^	*bla* _CTX-M-55_
SA296	Give	AMP-CRO-FEP-FFC-FOT-GEN-OXA-STR-TAZ	*bla* _CTX-M-55_	AMP-CRO-FEP-FFC-FOT-GEN-OXA-STR-TAZ	*bla* _CTX-M-55_
SA316	Sinstorf	AMP-CRO-FFC-FOT-GEN-OXA-STR-TAZ	*bla* _CTX-M-55_	AMP-CRO-FFC-FOT-GEN-OXA-STR-TAZ	*bla* _CTX-M-55_
SA342	Sinstorf	AMP-CRO-FFC-FOT-GEN-OXA-STR-TAZ	*bla* _CTX-M-55_	AMP-CRO-FFC-FOT-GEN-OXA-STR-TAZ	*bla* _CTX-M-55_
SA365	Sinstorf	AMP-CRO-FFC-FOT-GEN-OXA-STR-TAZ	*bla* _CTX-M-55_	AMP-AZI-CRO-FFC-FOT-GEN-OXA-STR-TAZ ^b^	*bla* _CTX-M-55_
SA398	Sinstorf	AMP-CRO-FFC-FOT-GEN-OXA-STR-TAZ	*bla* _CTX-M-55_	AMP-CRO-FFC-FOT-GEN-OXA-STR-TAZ	*bla* _CTX-M-55_
SA435	Sinstorf	AMP-CRO-FEP-FFC-FOT-GEN-OXA-STR-TAZ	*bla* _CTX-M-55_	AMP-CRO-FEP-FFC-FOT-GEN-OXA-STR-TAZ	*bla* _CTX-M-55_
SA467	Sinstorf	AMP-CRO-FFC-FOT-GEN-OXA-STR-TAZ	*bla* _CTX-M-55_	AMP-CRO-FFC-FOT-GEN-OXA-STR-TAZ	*bla* _CTX-M-55_
SA480	Sinstorf	AMP-CRO-FFC-FOT-OXA-STR	*bla* _CTX-M-55_	AMP-CRO-FFC-FOT-OXA-STR	*bla* _CTX-M-55_
SA519	Sinstorf	AMP-CRO-FFC-FOT-GEN-OXA-STR-TAZ	*bla* _CTX-M-55_	AMP-CRO-FFC-FOT-GEN-OXA-STR-TAZ	*bla* _CTX-M-55_

^a^ Abbreviation: AMP, ampicillin; AZI, azithromycin; CIP, ciprofloxacin; CRO, ceftriaxone; FEP, cefepime; FFC, florfenicol; FOT, cefotaxime; FOX, cefoxitin; GEN, gentamicin; NAL, nalidixic acid; OXA, oxacillin; STR, streptomycin; TAZ, ceftazidime. ^b^ Difference resistance profile observed of transconjugant compared to donor.

**Table 3 antibiotics-14-01198-t003:** Characteristics of MDR ESBL-producing and fluoroquinolone-resistant *Salmonella* isolates based on WGS analysis (*n* = 12).

ID.	Serovar ^c^	Resistance Pattern ^b^		AMR Gene	QRDR Mutations		Localization
		Determined by AST	Predicted by WGS		*gyrA*	*parC*	
SA69	Kentucky	AMP-CIP-CRO-FFC-FEP-FOT-GEN-LEV-NAL-OXA-STR-TAZ	AMK-AMX-AMP-APR-ATM-FEP-FOT-TAZ-CRO-CEP-CHL-DKB-DOX-FOT-GEN-MNO-NET-PIP-SIS-SPT-STR-TCY-TIC-TGC-TOB	*aadA2*, *aac(6′)-Iaa*, *aph(3″)-Ib*, *florR*, *bla*_TEM-1_, *bla*_CTX-M-55_, *tet*	-	-	pSA69-HI2/N
				-	G259A, C248T	G239T, C170G	chromosome
SA74	Kentucky	AMP-AZI-CIP-CRO-FFC-FEP-FOT-GEN-LEV-NAL-OXA-STR-TAZ	AMK-AMX-AMP-APR-ATM-FEP-FOT-TAZ-CRO-CEP-CHL-DKB-DOX-FOT-GEN-MNO-NET-PIP-SIS-SPT-STR-SMX-TCY-TIC-TGC-TOB-TMP	*florR*, *tet*, *bla*_TEM-1_, *bla*_CTX-M-55_, *aac(6′)-Iaa*, *aac(6′)-IIb*, *aph(3′)-Ib*, *sul*1, *dfrA*, *aadA*	-	-	pSA74-HI2/N
				-	G259A, C248T	G239T, C170G	chromosome
SA81	Derby	AMP-AZI-CIP-CRO-FFC-FOT-LEV-NAL-OXA-STR-TAZ	AMK-AMX-AMC-AMP-AMP+C-ATM-FEP-FOT-FOX-TAZ-CEP-CHL-CIP-DKB-DOX-FOT-FOS-GEN-KAN-NAL-NEO-NET-PAR-PIP-TZP-RST-SIS-STR-SMX-TCY-TIC-TCC-TOB	*florR*, *tet*, *bla*_VEB_, *bla*_TEM-1_, *sul*2, *aph(3′)-Ia*, *aac(6′)-IIa*,	-	-	pSA81-C
				-	C248T	C170G	chromosome
SA105	Stanley	AMP-CIP-CRO-FFC-FEP-FOT-GEN-NAL-OXA-STR-TAZ	AMK-AMX-AMP-APR-CEP-CHL-CIP-DKB-DOX-FOT-GEN-MNO-NET-PIP-SIS-SPT-STR-TCY-TIC-TOB-TMP	*qnrS*, *acc(3′)-IId*, *acc(6′)IIa*, *aadA*2, *tet*, *bla*_TEM-1_, *florR*, *dfrA*	-	-	*-*
				-	-	C170G	chromosome
SA149 ^a^	Wandsworth	AMP-AZI-CIP-CRO-FFC-FEP-FOT-GEN-NAL-OXA-STR-TAZ	AMK-AMP-AMP+C-FEP-FOT-TAZ-CHL-CIP-GEN-SMX-TCY-TGC-TOB-TMP	*acc(3′)-IId*, *acc(6′)Iaa*, *florR*, *bla*_TEM_, *dfrA*, *sul1*, *qnrS*, *tet*	-	-	pSA149-HI2/N
				-	-	C170G	chromosome
SA150 ^a^	Wandsworth	AMP-AZI-CIP-CRO-FFC-FEP-FOT-GEN-NAL-OXA-STR-TAZ	AMK-AMX-AMC-AMP-AMP+C-APR-ATM-FEP-FOT-TAZ-CRO-CEP-CHL-CIP-CLI-DKB-DOX-ERY-FOT-GEN-LIN-MNO-NET-PIP-TZP-PRI-QDA-SIS-SPT-STR-SMX-TCY-TIC-TCC-TGC-TOB-TMP	*aadA2*, *aph(6′)-Id*, *aac(3′)-IId*, *aac(6′)-Iaa*, *floR*, *bla*_CTX-M-55_, *sul1*, *dfrA*, *erm*, *qnrS*, *tet*	-	-	pSA150-HI2/N
				-	-	C170G	chromosome
SA153	Mons	AMP-AZI-CIP-CRO-FFC-FEP-FOT-FOX-GEN-NAL-OXA-STR-TAZ	AMK-AMX-AMC-AMP-AMP+C-APR-ATM-FEP-FOT-TAZ-CRO-CEP-CHL-CIP-CLI-DKB-DOX-ERY-FOT-GEN-LIN-MNO-NET-PIP-TZP-PRI-QDA-SIS-SPT-STR-SMX-TCY-TIC-TCC-TGC-TOB-TMP	*bla_TEM-1_*, *sul*, *dfrA*, *erm*, *aadA2*, *floR*, *qacE*, *erm*, *tet(M)*, *qnrS*, *aac(3′)-IId*, *aph(3′)-Ib*, *aac(3′)-Iaa*	-	-	pSA153-HI2/N
				-	-	C170G	chromosome
SA175	Typhimurium	AMP-CIP-CRO-FFC-FEP-FOT-GEN-LEV-NAL-OXA-STR-TAZ	AMK-AMX-AMP-APR-ATM-FEP-FOT-TAZ-CRO-CEP-CHL-CIP-COL-DKB-DOX-FOT-GEN-NAL-NET-PIP-SIS-STR-SMX-TCY-TIC-TOB	*mcr*, *tet*, *bla*_CTX-M-55_, *qepA*, *qnrS*, *catA*, *sul*, *floR*, *aac(3′)IId*, *aph(3′)Ib*, *aac(6′)Iaa*	-	-	pSA175-C
				*bla* _TEM-1_	-	-	pSA175-N
				-	C248A	-	chromosome
SA231	Kentucky	AMP-CIP-CRO-FFC-FEP-FOT-GEN-LEV-NAL-OXA-STR-TAZ	AMK-AMX-AMP-APR-ATM-FEP-FOT-TAZ-CRO-CEP-CHL-CIP-DKB-DOX-FOT-GEN-MNO-NAL-NET-PIP-SIS-SPT-STR-SMX-TCY-TIC-TGC-TOB-TMP	*bla*_TEM-1_, *bla*_CTX-M-55_, *floR*, *qacE*, *sul1*, *dfrA*, *aadA2*, *aac(3)IId*, *aadA2*, *aph(3′)Ib*, *aac(6′)Iaa*, *tet.*	-	-	pSA231-HI2/N
				-	G259A, C248T	G239T, C170G	chromosome
SA510	Agona	AMP-CIP-CRO-FFC-FOT-FOX-GEN-LEV-NAL-OXA-STR-TAZ	AMK-AMX-AMC-AMP-AMP+C-FOT-FOX-TAZ-CEP-CHL-CIP-DKB-DOX-FOT-FOR-FOS-GEN-KAN-LIN-NAL-NEO-NET-PIP-TZP-SIS-SPT-STR-SMX-TCY-TIC-TCC-TOB	*aph(3′)Ia*, *aac(6′)Ib*, *aac(3)*VIa, *aph(6)Id*, *tet*, *sul*2, *floR*, *bla*_TEM-1_, *bla*_CMY_, *fosA*, *aadA*1, *Inu*	-	-	pSA510-C
				*-*	-	-	pSA510-F
				-	C248T	C170G	chromosome
SA523	Agona	AMP-AZI-CIP-CRO-FFC-FOT-FOX-GEN-LEV-NAL-OXA-STR-TAZ	AMK-AMX-AMC-AMP-AMP+C-FOT-FOX-TAZ-CEP-CHL-CIP-DKB-DOX-FOT-FOR-FOS-GEN-KAN-LIN-NAL-NEO-NET-PIP-TZP-SIS-SPT-STR-SMX-TCY-TIC-TCC-TOB	*Inu*, *sul*2, *aac(6′)Ib*, *aph(3′)Ia*, *aadA*1, *aac(3)*VIa, *bla*_TEM-1_, *bla*_CMY_, *qnrS*, *fosA*, *floR*, *tet*	-	-	pSA523-C
				*-*	-	-	pSA523-F
				-	C248T	C170G	chromosome
SA485	Agona ^c^	AMP-AZI-CIP-FFC-FOT-FOX-GEN-LEV-NAL-OXA-STR-TAZ	AMK-AMX-AMC-AMP-AMP+C-FOT-FOX-TAZ-CEP-CHL-CIP-DKB-DOX-FOT-FOR-FOS-GEN-KAN-LIN-NAL-NEO-NET-PIP-TZP-SIS-SPT-STR-SMX-TCY-TIC-TCC-TOB	*floR*, *aac(6′)Ib*, *aadA*1, *sul*2, *tet*, *bla_CMY_*, *bla_TEM-_*_1_, *fosA*, *Inu*, *aac(6′)Iaa*, *aac(3)*VIa, *aac(6′)Ib*, *aph(3′)Ia*	-	-	pSA485-C
				-	-	-	pSA485-F
				-	C248T	C170G	chromosome

^a^ Capable of conjugative transfer, ^b^ Abbreviation: AMR, amikacin; AMX, amoxicillin; AMC, amoxicillin+clavulanic acid; AMP, ampicillin; AMP+C, ampicillin/clavulanic acid; APR, apramycin; ATM, Aztreonam; AZI, azithromycin; CEP, cephalothin; CHL, chloramphenicol; CIP, ciprofloxacin; CRO, ceftriaxone; DKB, dibekacin; DOX, doxycycline; FEP, cefepime; FFC, florfenicol; FOT, cefotaxime; FOS, fosfomycin; FOX, cefoxitin; GEN, gentamicin; KAN, kanamycin; LIN, lincomycin; MNO, minocycline; NAL, nalidixic acid; NEO, neomycin; NET, netilmicin; OXA, oxacillin; PIP, piperacillin; QSA, Quinupristin; RST, ribostamycin; SIS, sisomicin; SMX, sulfamethoxazole; SPT, spectinomycin; STR, streptomycin; TAZ, ceftazidime; TCY, tetracycline; TCC, ticarcillin/clavulanic acid; TIC, ticarcillin; TGC, tigecycline; TMP, trimethoprim; TOB, tobramycin; TZP, piperacillin/tazobactam. ^c^ Non-ESBL producer for comparison.

**Table 4 antibiotics-14-01198-t004:** Probability of exposure, probability of illness, and annual risk of illness per person.

Statistics	Probability of Exposure (Pe)				Probability of Illness (Pi)				Annual Risk of Illness per Person			
	Cooking Time at 64 °C											
	Uncooked	1 min ^b^	2 min	3 min	Uncooked	1 min	2 min	3 min	Uncooked	1 min	2 min	3 min
Scenario 1: ESBL-producing *Salmonella*												
Mean	9.15 × 10^−1 a^	1.98 × 10^−1^	7.96 × 10^−3^	6.65 × 10^−5^	5.06 × 10^−2^	7.61 × 10^−4^	2.09 × 10^−5^	1.77 × 10^−7^	8.85 × 10^−1^	9.49 × 10^−2^	2.52 × 10^−4^	1.77 × 10^−8^
Std Dev	2.08 × 10^−1^	2.23 × 10^−1^	1.34 × 10^−2^	1.11 × 10^−4^	5.29 × 10^−2^	1.29 × 10^−3^	3.57 × 10^−5^	3.15 × 10^−7^	2.61 × 10^−1^	1.842 × 10^−1^	1.29 × 10^−3^	8.90 × 10^−8^
Min ^c^	4.19 × 10^−5^	1.28 × 10^−5^	2.13 × 10^−7^	1.55 × 10^−9^	9.00 × 10^−7^	6.80 × 10^−9^	6.57 × 10^−11^	1.31 × 10^−13^	9.59 × 10^−8^	1.52 × 10^−10^	0.00 × 10^0^	0.00 × 10^0^
Max ^d^	1	9.99 × 10^−1^	2.17 × 10^−1^	1.82 × 10^−3^	3.57 × 10^−1^	2.58 × 10^−2^	5.44 × 10^−4^	5.71 × 10^−6^	0	9.98 × 10^−1^	4.00 × 10^−2^	2.86 × 10^−6^
Scenario 2: Ciprofloxacin-resistant *Salmonella*												
Mean	7.11 × 10^−1^	3.99 × 10^−2^	8.16 × 10^−3^	3.01 × 10^−4^	1.15 × 10^−2^	1.15 × 10^−4^	2.12 × 10^−5^	7.44 × 10^−7^	6.35 × 10^−1^	5.70 × 10^−3^	2.29 × 10^−4^	3.25 × 10^−7^
Std Dev	3.31 × 10^−1^	5.96 × 10^−2^	1.39 × 10^−2^	5.30 × 10^−4^	1.64 × 10^−2^	2.07 × 10^−4^	3.78 × 10^−5^	1.67 × 10^−11^	3.88 × 10^−1^	2.31 × 10^−2^	1.09 × 10^−3^	1.50 × 10^−6^
Min	6.22 × 10^−5^	2.64 × 10^−7^	2.38 × 10^−8^	2.96 × 10^−9^	2.80 × 10^−8^	4.20 × 10^−9^	1.24 × 10^−10^	1.67 × 10^−11^	1.99 × 10^−10^	8.10 × 10^−14^	0.00 × 10^0^	0.00 × 10^0^
Max	1	6.79 × 10^−1^	2.55 × 10^−1^	7.63 × 10^−3^	1.88 × 10^−1^	4.83 × 10^−3^	7.06 × 10^−4^	2.29 × 10^−5^	0	5.19 × 10^−1^	4.40 × 10^−2^	4.38 × 10^−5^
Scenario 3: *Salmonella*												
Mean	9.13 × 10^−1^	4.03 × 10^−1^	7.51 × 10^−3^	6.41 × 10^−5^	5.10 × 10^−2^	7.62 × 10^−4^	1.96 × 10^−5^	1.64 × 10^−7^	9.53 × 10^−1^	2.82 × 10^−1^	2.15 × 10^−4^	1.59 × 10^−8^
Std Dev	2.10 × 10^−1^	3.25 × 10^−1^	1.24 × 10^−2^	1.15 × 10^−4^	5.37 × 10^−2^	1.31 × 10^−3^	3.41 × 10^−5^	2.80 × 10^−7^	1.71 × 10^−1^	3.32 × 10^−1^	1.13 × 10^−3^	1.13 × 10^−7^
Min	8.99 × 10^−4^	2.02 × 10^−6^	8.25 × 10^−8^	3.03 × 10^−10^	8.57 × 10^−7^	3.42 × 10^−9^	1.57 × 10^−11^	1.82 × 10^−13^	6.22 × 10^−6^	2.08 × 10^−11^	0	0
Max	1	1	2.00 × 10^−1^	2.35 × 10^−3^	3.39 × 10^−1^	2.16 × 10^−2^	5.37 × 10^−4^	4.95 × 10^−6^	1	1	5.27 × 10^−2^	7.06 × 10^−1^

^a^ E, exponent notation representing ×10^. ^b^ min, minute. ^c^ min, minimum ^d^ max, maximum.

**Table 5 antibiotics-14-01198-t005:** Sensitivity correlation or regression values of risk model.

Conditions (Cooking Time/min ^c^)	Regression (R)	
	Concentration (CFU/g) ^a^	Consumption (g/day) ^b^
Scenario 1 ESBL-producing *Salmonella*		
uncooked	0.306	0.309
1 min	0.584	0.593
2 min	0.383	0.380
3 min	0.393	0.375
Scenario 2 Ciprofloxacin-resistant *Salmonella*		
uncooked	0.488	0.475
1 min	0.424	0.435
2 min	0.383	0.382
3 min	0.399	0.404
Scenario 3 *Salmonella*		
uncooked	0.488	0.475
1 min	0.424	0.435
2 min	0.383	0.382
3 min	0.399	0.404

^a^ CFU/g, colony-forming unit/gram, ^b^ g/day, gram per day, ^c^ min, minute.

**Table 6 antibiotics-14-01198-t006:** Primers used in this study.

Targets	Primers	Amplicon Size (bp)	Primer Sequence (5′–3′)	References
QRDR				
*gyrA*	gyrA-F	436	GCTGAAGAGCTCCTATCTGG	Chuanchuen and Padungtod [63]
	gyrA-R		GGTCGGCATGACGTCCGG	
*parC*	parC-F	390	GTACGTGATCATGGATCGTG	Chuanchuen and Padungtod [63]
	parC-R		TTCCTGCATGGTGCCGTCG	
PMQR				
*qnrA*	qnrA-F	516	ATTTCTCACGCCAGGATTTG	Stephenson, et al. [64]
	qnrA-R		GATCGGCAAAGGTTAGGTCA	
*qnrB*	qnrB-F	469	GATCGTGAAAGCCAGAAAGG	Stephenson, Brown, Holness and Wilks [64]
	qnrB-R		ACGATGCCTGGTAGTTGTCC	
*qnrS*	*qnrS*-F	417	ACGACATTCGTCAACTGCAA	Stephenson, Brown, Holness and Wilks [64]
	*qnrS*-R		TAAATTGGCACCCTGTAGGC	
*qepA*	*qepA*-F	199	GCAGGTCCAGCAGCGGGTAG	Yamane, et al. [65]
	*qepA*-R		CTTCCTGCCCGAGTATCGTG	
*aac(6′)-Ib-cr*	*aac(6′)*-Ib-F	482	TTGCGATGCTCTATGAGTGGCTA	Park, et al. [66]
	*aac(6′)*-Ib-R		CTCGAATGCCTGGCGTGTTT	
ESBLs				
*bla* _CTX-M_	*bla*CTX-M-F	585	CGATGTGCAGTACCAGTAA	Batchelor, et al. [67]
	*bla*CTX-M-R		AGTGACCAGAATCAGCGG	
*bla* _TEM_	*bla*TEM-F	964	GCGGAACCCCTATTT	Hasman, et al. [68]
	*bla*TEM-R		TCTAAAGTATATATGAGTAAACTTGGTCT	
*bla* _SHV_	*bla*_SHV_-F	854	TTCGCCTGTGTATTATCTCCCTG	Hasman, Mevius, Veldman, Olesen and Aarestrup [68]
	*bla*_SHV_-R		TTAGCGTTGCCAGTGYTG	
*bla*_CTXM_ subgroup 1 ^a^	CTXMgr1-F	688	TTAGGAARTGTGCCGCTGYA ^b^	Dallenne, et al. [69]
	CTXMgr1-R		CGATATCGTTGGTGGTRCCAT ^b^	
*bla*_CTXM_ subgroup 2 ^c^	CTXMgr2-F	404	CGTTAACGGCACGATGAC	Dallenne, Da Costa, Decre, Favier and Arlet [69]
	CTXMgr2-R		CGATATCGTTGGTGGTRCCAT ^b^	
*bla*_CTXM_ subgroup 8/25 ^d^	CTX-M gr.8/25-F	326	AACRCRCAGACGCTCTAC ^b^	Dallenne, Da Costa, Decre, Favier and Arlet [69]
	CTX-M gr.8/25-R		TCGAGCCGGAASGTGTYAT ^b^	
*bla*_CTXM_ subgroup 9 ^e^	CTX-M gr.9-F	850	GTGACAAAGAGAGTGCAACGG	Sabaté, et al. [70]
	CTX-M gr.9-R		ATGATTCTCGCCGCTGAAGCC	

^a^ Group 1 includes *bla*_CTX-M-1, -3, -10 to -12, -15, -22, -23, -28, -29, and -30._
^b^ Y = T or C; R = A or G; S = G or C; D = A, G, or T. ^c^ Group 2 includes *bla*_CTX-M-2, -4 to -7, and -20._ ^d^ Group 8/25 includes *bla*_CTX-M-8, CTX-M-25, CTX-M-26, and CTX-M-39 to CTX-M-41._ ^e^ Group 9 includes *bla*_CTX-M-9, -14, -16 to -19, -21, and -27._

**Table 7 antibiotics-14-01198-t007:** Risk assessment equations.

Eq.	Equation	Description
(1)	$LR=\frac{t}{DR_{64}}$	LR: log reduction t: cooking time at 0, 1, 2, and 3 min DR64: decimal reduction time at 64 °C for *Salmonella* with the value of 0.48
(2)	$RCc=Cc\;\times \;10^{-LR}$	RCc: remaining concentration of *Salmonella* in pork after cooking at 64 °C Cc: concentration of *Salmonella* in pork (CFU/gram)
(3)	D=RCc × Cs	D: dose of ingestion (CFU/day) Cs: daily consumption of pork
(4)	$Pe=1-e^{-D}$	Pe: probability of consuming at least 1 CFU of *Salmonella* from pork e: Euler’s constant = 2.71828182845904
(5)	$P_{i}=1-{\left(1+\left(\frac{D}{51.45}\right)\right)}^{-0.1324}$	Pi: probability of illness
(6)	$Pi_{day}=Pi\;\times \;Pe$	Pi day: probability of illness per day
(7)	$Pi_{year}=1-{\left(1-Pi_{day}\right)}^{365}$	Pi year: probability of illness per year

## Data Availability

The genome sequencing and assembly data in this study was available in NCBI database The BioProject number PRJNA946550.

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
