# Peer review of "Integrated Genetic Characterization and Quantitative Risk Assessment of Cephalosporin- and Ciprofloxacin-Resistant Salmonella in Pork from Thailand"

_antibiotics, 2025, doi:10.3390/antibiotics14121198_

Round 1

Reviewer 1 Report

Comments and Suggestions for Authors

The present manuscript aimed to assess risk associated with third-generation cephalosporin and fluoroquinolone-resistant Salmonella from pork consumption by integrating phenotypic resistance profiles with genetic data to characterize the risks and transmission pathways. This paper is of high public health relevance; however, several areas of concern must be addressed to improve the quality of this manuscript.

Areas of concern:

Abstract

Lines 21-22: Please, consider briefly mentioning the Salmonella isolation method from sample

Lines 39: The following sentence “Thorough cooking…..reduction” is incomplete; please consider combining lines 39 and 40 to make a complete sentence.

Lines 39-41: Please, consider adding a statement for a recommendation for practice or policy

  1. Introduction

Lines 50-53: Please, consider adding a transitional statement to introduce “Antimicrobial resistance” that is well described in paragraph 2.

  1. Results

Line 106: Please, consider consistently using “MPN/g” and not “g MPN”

Lines 214 and 420: Please, consider italicizing “E.coli”

  1. Discussion

Line 350: Please, add “s” to “observation”

Lines 373-386: There is need for a logical flow between AMR patterns, genetic findings (HGT/VGT), and risk model results; moreover, there is also need to concisely link your AMR findings to risk results or risk outcomes.

Lines 468-473: Although the novelty of your risk model was clearly highlighted, however, you failed to recommend future study for its validation in the conclusion section.

  1. Materials and Methods

Sampling Design and Sample Collection

Lines 495-497: The phrase “proportionally selected” is ambiguous; please, consider specifying the basis of proportionality (population density or number of markets?).

Line 520: Please, consider specifying the exact version of Jarvis’s software used.

Line 535: Please, consider replacing “DifcoTM” with the correct brand name (BD Difco™).

Lines 548-549: Please, consider briefly adding PCR reaction conditions (annealing temperature, cycles, etc).

Lines 579-580: For clarity and better flow, consider replacing “Transfer of plasmid carrying ESBL gene, was assessed using Biparental filter mating method” with “Plasmid transfer was evaluated by biparental filter mating following standard protocols.” Additionally, “Biparental” should be not written with capital B.

Lines 587-620: For better flow, consider subtitling each paragraph under section 4.7

Author Response

Comments 1: Lines 21-22: Please, consider briefly mentioning the Salmonella isolation method from sample

Response 1: The sentences are now revise to included the Salmonella isolation method as suggested. Please see line 21-23

Comments 2: Lines 39: The following sentence “Thorough cooking…..reduction” is incomplete; please consider combining lines 39 and 40 to make a complete sentence.

Response 2: The sentences are now revised for better reading and understanding. Please see line 39-41.

Comments 3: Lines 39-41: Please, consider adding a statement for a recommendation for practice or policy

Response3: The statement for a recommendation for practice or policy is now included as recommended. Please see line 41-43

Comments 4: Lines 50-53: Please, consider adding a transitional statement to introduce “Antimicrobial resistance” that is well described in paragraph 2.

Response 4: The sentence to introduce antimicrobial resistance is now added. Please see line 57-59.

Comments 5: Line 106: Please, consider consistently using “MPN/g” and not “g MPN”

Response 5: The MPN/g were now corrected, and the other typographical error has also been fixed. Please see line 111-112.

Comments 6: Lines 214 and 420: Please, consider italicizing “E. coli”

Response 6: The “E. coli” is now italicized. Please see line 238 and 440.

Comments 7: Line 350: Please, add “s” to “observation”

Response 7: The “s” is now added to read “observations”. Please see line 370.

Comments 8: Lines 468-473: Although the novelty of your risk model was clearly highlighted, however, you failed to recommend future study for its validation in the conclusion section.

Response 8: Ways to improve the model validation are added for better understanding (line 493-500)

Comments 9: Lines 495-497: The phrase “proportionally selected” is ambiguous; please, consider specifying the basis of proportionality (population density or number of markets?).

Response 9: The words “based on population density” are now added for better clarification. Please see line 522-523.

Comments 10: Line 520: Please, consider specifying the exact version of Jarvis’s software used.

Response 10: The version 6 of Jarvis’s software was used and is now added as suggested. Please see line 547.

Comments 11: Line 535: Please, consider replacing “DifcoTM” with the correct brand name (BD Difco™).

Response 11: The DifcoTM is now replaced with “BD Difco™” as suggested. Please see line 562.

Comments 12: Lines 548-549: Please, consider briefly adding PCR reaction conditions (annealing temperature, cycles, etc).

Response 12: All PCR reaction conditions used are now included. Please see line 575-585.

Comments 13: Lines 579-580: For clarity and better flow, consider replacing “Transfer of plasmid carrying ESBL gene, was assessed using Biparental filter mating method” with “Plasmid transfer was evaluated by biparental filter mating following standard protocols. Additionally, “Biparental” should be not written with capital B.

Response 13: The sentence was corrected for better understanding as suggested. Please see line 619-620.

Comments 14: Lines 587-620: For better flow, consider subtitling each paragraph under section 4.7

Response 14: All four paragraphs in section is now 4.7 are now subtitled as suggested. Please see section 4.7. on page 21 (line 627, 633 and 646).

Reviewer 2 Report

Comments and Suggestions for Authors

The manuscript written by Thawanrut Kiatyingangsulee, Si Thu Hein Rangsiya Prathan, Songsak Srisanga, Saharuetai Jeamsripong and Rungtip Chuanchuen entitled „ Integrated Genetic Characterization and Quantitative Risk Assessment of Cephalosporin- and Ciprofloxacin-Resistant Salmonella in pork from Thailand” is a complex study that enriches the current knowledge regarding the phenotypic resistance of Salmonella strains isolated from pork meat in Thailand.

The manuscript has several writing errors that have to be resolved:

L 21 Please replace n=000 with corresponding value

L21-24 Please use full description when first mentioned in text (ESBL-producing ,Quantitative AMR risk assessment).

L 38 The authors mention that „raw pork consumption represents the highest risk”. Is this a current practice in Thailand, the consumption of raw pork meat? I believe a small introduction regarding the culinary preferences in Thailand and the specific raw pork products would help the readers understand better the situation, and why raw pork was chosen as one of the scenarios.  Also, please change „raw pork” with „raw pork meat” in all the text.

L 99 Values could be expressed with a 95% confidence interval.

L 148 Table 2 please add full description of the abbreviated terms in the legend. Also, Table 5, Please check for similar errors in the text.

L 214 Please verify that all species names are written in italics.

Figure 4 is before Figure 3 please add tables and figures in corresponding order and close to the paragraph where they are mentioned.

L 276, L280 Please verify all concentrations. The first concentration mentioned is 3.34, is it correct?

L 308-311 This paragraph is a bit confusing, please try to rephrase it.

Table 1 appears in materials and methods section which subchapter 4, please add corresponding number according to the order in the text.

L642 Revise the numbers of the chapters, I believe this should be 4.8.2 not 2.8.2

Author Response

Comments 1: L 21 Please replace n=000 with corresponding value

Response 1: The sentences are revised, and the exact number of isolates are also included. Please see line 23 and 24.

Comments 2: L21-24 Please use full description when first mentioned in text (ESBL-producing, Quantitative AMR risk assessment).

Response 2:  The full descriptions are included for ESBL and AMR. Please see lines 23 and 25.

Comments 3: L 38 The authors mention that „raw pork consumption represents the highest risk”. Is this a current practice in Thailand, the consumption of raw pork meat? I believe a small introduction regarding the culinary preferences in Thailand and the specific raw pork products would help the readers understand better the situation, and why raw pork was chosen as one of the scenarios.  Also, please change „raw pork” with „raw pork meat” in all the text.

Response 3: The consumption of raw pork meat is common in certain regions of Thailand. The reasons for choosing raw pork meat in this study are now included for better clarification. Please see line 54-57.

The “raw pork meat” is now used throughout the text. Please see line 21, 39, 54, 318, 399, 407, 408, 508, 512 and 693.

Comments 4: L 99 Values could be expressed with a 95% confidence interval.

Response 4: The authors understand that L99 here refers to the result section. If so, most of phenotypic and genotypic values are based on descriptive statistics, of which 95% confidence interval is not required. However, “at 95% confidence interval” are now added in the result part of risk assessment for better clarification. Please see line 296, 303 and 315.

Comments 5: L 148 Table 2 please add full description of the abbreviated terms in the legend. Also, Table 5, Please check for similar errors in the text.

Response 5:  The full description of the abbreviations is now added in the legend of Table 1 (formerly Table 2) and Table 4 (formerly Table 5). Please see Table 1, line 155-157 and Table 4, line 329-332.

Comments 6: L 214 Please verify that all species names are written in italics.

Response 6: All bacterial species are checked throughout the manuscript and corrected to be in italics.

Comments 7: Figure 4 is before Figure 3 please add tables and figures in corresponding order and close to the paragraph where they are mentioned.

Response 7: All tables and figures are re-numbered and placed close to the paragraph where they are mentioned. Please see Figure 1, line 135; Figure 2, line 225; Figure 3, line 248; Figure 4, line 272; Figure 5, line 353; and Figure 6, line 681. Tables are also re-organized. Please see response 10.

Comments 8:L 276, L280 Please verify all concentrations. The first concentration mentioned is 3.34, is it correct?

Response 8: The first concentration of 3.34 is correct. All concentrations were verified and confirmed as accurate (line 288, 290 and 292).

Comments 9: L 308-311 This paragraph is a bit confusing, please try to rephrase it.

Response 9: The paragraph is now revised for better understanding. Please see line 321-326.

Comments 10: Table 1 appears in materials and methods section which subchapter 4, please add corresponding number according to the order in the text.

Response 10: All tables were checked and re-ordered as suggested. Please see Table 1, line 154; Table 2, line 202; Table 3, line 207; Table 4, line 328; Table 5, line 358; Table 6, line 594 and Table 7, line 678. 

Comments 11:   L642 Revise the numbers of the chapters, I believe this should be 4.8.2 not 2.8.2

Response 11: The chapter number is now updated from 4.8.2. to 4.8.6. Please see line 685, 716, 721, 729 and 734.

Round 2

Reviewer 1 Report

Comments and Suggestions for Authors

The authors have carefully addressed all the reviewer's comments. This has positively impacted the quality of the manuscript.

Author Response

Comment: The authors have carefully addressed all the reviewer's comments. This has positively impacted the quality of the manuscript.
Response: Thank you very much for your positive feedback.

Reviewer 2 Report

Comments and Suggestions for Authors

The authors have made the corresponding changes. 

L288 If value 3,34 is correct please write it as 3,34 x 10-1 for consistency. Same thing for line 292

Author Response

Comment 1: L288 If value 3.34 is correct please write it as 3.34 x 10-1 for consistency. Same thing for line 292.

Response 1: The value 3.34 is correct. However, the authors believe that reporting the concentration with a single leading digit and two decimal places is appropriate and consistent with standard practice. Presenting the value in decimal form makes it easier to interpret, whereas scientific notation is typically reserved for values that are very large or small. For consistency, we have corrected 11.11 to 1.11 × 10¹, using a single leading digit. Please see line 292.